# What are the perceived target groups and occasions for wines and beers labelled with verbal and numerical descriptors of lower alcohol strength? An experimental study

Milica Vasiljevic,[1,2] Dominique-Laurent Couturier,[1] Theresa M Marteau[1]

[1]Behaviour and Health Research Unit, Institute of Public Health, University of Cambridge, Cambridge, UK
[2]Department of Psychology, Durham University, Durham, UK

**Correspondence to**
Dr. Milica Vasiljevic;
milica.vasiljevic@medschl.cam.ac.uk;
milica.vasiljevic@durham.ac.uk

## ABSTRACT

**Objectives** Alcohol consumption is the fifth leading cause of morbidity and mortality globally. The development and promotion of lower strength alcohol products may help reduce alcohol consumption and associated harms. This study assessed what a sample of UK weekly drinkers perceived to be the target groups and occasions for drinking wines and beers labelled with different verbal and numerical descriptors of lower alcohol strength.

**Design and participants** 3390 adults (1697 wine and 1693 beer drinkers) were sampled from a nationally representative UK panel, and participated in a between-subjects experiment in which participants were randomised to 1 of 18 groups with one of three levels of verbal descriptor (*Low* vs. *Super Low* vs. *No verbal descriptor*) and six levels of %ABV (five levels varying for wine and beer, and no level given).

**Measures** The study gauged participants' perceptions of the type of person that would find the randomised beverage appealing and the type of occasion on which the beverage is likely to be drunk at.

**Results** A principal component analysis showed that participants perceived pregnant women, sportspeople and those aged 6–13 years old were the target groups for products labelled with 0%ABV or the verbal descriptors *Low* or *Super Low*, whereas men, women, and those aged above 18 were perceived as the target groups for products labelled with higher %ABV. Participants also rated the products labelled with 0%ABV or the verbal descriptors *Low* or *Super Low* as targeting consumption on weekday lunches, whereas products labelled with higher %ABV were rated as targeting dinner/evening occasions, including parties, holidays and celebrations.

**Conclusions** Lower strength products were seen as targeting non-traditional consumers (pregnant women) and occasions (weekday lunchtimes), suggesting these products may be perceived as extensions to regular strength alcoholic drinks rather than as substitutes for them.

## BACKGROUND

Globally alcohol consumption is the fifth leading cause of morbidity and mortality.[1]

### Strengths and limitations of this study

► A notable strength of the present experimental study is the large sample of weekly wine and beer drinkers drawn from a nationally representative panel of the UK population.
► The use of fictitious non-branded labels further strengthens the conclusions that can be drawn from the present study, since we could rule out the confounding influence of brand recognition and loyalty.
► The present research is limited in using an online sampling frame with no behavioural outcome measures.

In the UK, the annual cost of alcohol-related harms has been estimated at £21 billion.[2] Although current figures show sales are dominated by regular (average) strength products on the market (UK: 12.9% for wine and 4.2% for beer),[3 4] there is a growing trend for consumers to more often buy lower strength and no-alcohol products.[5 6] Both industry and policy-makers have suggested that one way to address alcohol-related harms is the further development and greater availability of lower strength alcohol products (ie, products containing lower than average alcohol by volume [%ABV]).[7 8] The UK Government Alcohol Strategy published in March 2012 explicitly included an industry pledge through the Responsibility Deal to take one billion units of alcohol out of the market by 2015, mainly via greater consumer selection of lower strength alcohol products.[7]

One facet of increasing consumer selection of lower strength alcohol products includes the explicit labelling of lower strength alcoholic beverages. Low alcohol labels are a set of labels that carry descriptors such as 'low' or 'lighter' to denote low or reduced strength alcohol content in alcoholic drinks. Current

**Table 1** Participant demographic characteristics.

| Characteristic | Drink | |
| --- | --- | --- |
| | Wine (n=1697) | Beer (n=1693) |
| **Gender** | | |
| Male | 611 (36) | 1262 (75) |
| Female | 1086 (64) | 431 (25) |
| **Age group** | | |
| 18–35 | 207 (12) | 253 (15) |
| 36–45 | 295 (18) | 308 (18) |
| 46–60 | 560 (33) | 641 (38) |
| 61–99 | 635 (37) | 491 (29) |
| **Education[1]** | | |
| 4 GCSEs | 255 (15) | 341 (20) |
| 1 A-level | 310 (18) | 285 (17) |
| 2+A-levels | 287 (17) | 305 (18) |
| University | 781 (46) | 688 (41) |
| N/A | 64 (4) | 74 (4) |
| **Income[2]** | | |
| 0–15.5K pa | 306 (18) | 358 (21) |
| 15.51–25.5K pa | 290 (17) | 301 (18) |
| 25.51–40K pa | 499 (30) | 446 (26) |
| >40.01K pa | 497 (29) | 500 (30) |
| N/A | 105 (6) | 88 (5) |
| **Social grade** | | |
| Low | 167 (10) | 165 (10) |
| Medium | 328 (19) | 303 (18) |
| High | 203 (12) | 172 (10) |
| N/A | 999 (59) | 1053 (62) |
| **Index of Multiple Deprivation (IMD)[3]** | | |
| Quintile 1 | 230 (14) | 284 (17) |
| Quintile 2 | 263 (15) | 280 (16) |
| Quintile 3 | 307 (18) | 267 (16) |
| Quintile 4 | 268 (16) | 250 (15) |
| Quintile 5 | 271 (16) | 267 (16) |
| N/A | 358 (21) | 345 (20) |
| **Ethnicity** | | |
| White | 1592 (94) | 1580 (93.5) |
| Other | 97 (5.6) | 104 (6) |
| N/A | 8 (0.4) | 9 (0.5) |
| **Riskier drinkers** | | |
| No | 997 (58.8) | 750 (44) |
| Yes | 695 (41) | 942 (55.9) |
| N/A | 5 (0.2) | 1 (0.1) |

Note: Percentages appear in parentheses.
1GCSEs (General Certificate of Secondary Education) are usually taken at age 15–16 in the UK; A-Levels at age 17–18.
2Income bands are expressed per annum.
3Index of Multiple Deprivation (IMD) denotes neighbourhood-level deprivation; Quintile 1 reflects the highest level of deprivation and Quintile 5 the lowest level of deprivation.

European Union (EU) legislation limits the number of terms that can be used and further restricts the use of such descriptors to drinks of 1.2% alcohol by volume (ABV) and lower.[9] Globally similar restrictions apply.[10 11]

With a sunset clause ending in 2018, the UK's national regulations regarding the use of lower strength alcohol terms were repealed in December 2014. This provides an opportunity to consider revisions to the legislation including: (1) increasing the number of lower strength verbal descriptors; and (2) increasing the legislated strength limit above the current legislated cap of 1.2% ABV. The potential of lower strength alcohols to reduce consumption depends on whether: (1) lower strength products are consumed instead of higher strength products as opposed to simply increasing and extending the number of occasions perceived suitable for consuming alcohol,[12 13] and (2) lower strength products not leading to self-licensing, resulting in the higher overall consumption of alcohol than would have been consumed from a higher strength product alone.[14 15] However, the empirical evidence base regarding how consumers perceive and respond to products labelled as lower in alcohol strength is currently limited.

A recent study examining wine and beer consumption in a bar lab found that participants sampled from the general population of England drank about 20% more wine and beer when it was labelled as lower in alcohol strength, suggesting that lower strength alcohols may engender paradoxical effects.[16] Another recent study analysed the content of marketing messages on producers' and retailers' websites for lower and regular strength wines and beers sold online by the four main supermarkets in the UK.[17] This study found that lower strength alcohols were marketed not as substitutes for higher strength products but as ones that can be consumed on additional occasions.[17] Furthermore, lower strength wines and beers were more often marketed with claims to health benefits. These findings raise a broader question of the extent to which lower strength alcohol products will contribute to a public health strategy to reduce alcohol consumption and associated harms. However, this study only focused on the content of the marketing messages, and did not examine how consumers perceive and respond to alcohol products labelled as lower in strength.

To our knowledge, only one study to date has examined consumers' perceptions of strength (%ABV) and appeal of alcohol products described with lower alcohol verbal descriptors.[18] A sample of 1600 UK weekly wine and beer drinkers rated verbal descriptors *Low, Lower, Light, Lighter* and *Reduced* as denoting lower strength products than *Regular*, but higher strength than the cluster of lower strength verbal descriptors with intensifiers consisting of *Extra Low, Super Low, Extra Light* and *Super Light*. Among the two clusters of verbal descriptors *Low* and *Super Low* were the most differentiated descriptors. Drinks labelled with the verbal descriptor *Regular* (denoting average strength on the market) were rated as most appealing, with drinks

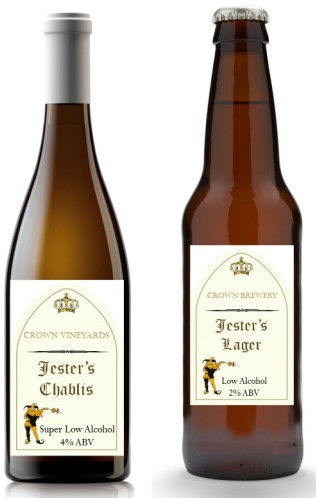

**Figure 1** Sample of two lower strength alcohol labels seen by participants (one in wine, one in beer).

labelled with the lower strength verbal descriptors using intensifiers rated least appealing.

The study was limited because the verbal descriptors of strength were not coupled with numerical information of percentage alcohol by volume (%ABV) on actual labels. Furthermore, the study only assessed participants' perceptions of strength and appeal of the different verbal descriptors. The current study aimed to fill this gap by combining numerical indicators of %ABV with a selection of verbal descriptors identified as most differentiating and understandable in the study by Vasiljevic and colleagues.[18] For the current study, we developed purposefully tailored labels so that we could control for any extraneous influences of participants' prior brand preference. The current study also extended previous studies by examining what weekly drinkers of wine and beer perceived to be the target groups and occasions of drinks labelled as lower in alcohol strength. Since this is the first study to examine weekly drinkers' perceptions of the target groups and occasions of drinks labelled with verbal and numerical information of lower alcohol strength, we did not have any prespecified hypotheses.

## METHOD
### Participants
Three thousand three hundred and ninety participants (1697 wine and 1693 beer drinkers) completed the study. Participants were sampled from an existing panel that was nationally representative for age, gender, SES and geographical region of the UK population. Of those who accessed the study link, only those who reported drinking alcohol at least once a week were eligible to continue further. Allocation to the wine or beer survey was based on drinking preference (see also Procedure). Since effect size estimates were not available for the outcomes of interest, the sample size calculations were based on differences in ratings of appeal of different

wines and beers labelled with verbal and/or numerical descriptors of lower alcohol strength observed in a pilot study. Attention checks were employed to screen out inattentive responders. Participants were informed in the Information Sheet and Consent Form that there would be attention checks in the online survey, and that failure to complete the attention checks correctly would result in them being prevented from completing the study. Attention was gauged by two items: *When was the last time you have flown to Mars? Please answer honestly and to the best of your knowledge: Never/A few days ago/ Weeks ago/Months ago.* Participants who did not choose the only plausible option of 'Never' were considered inattentive and were prevented from continuing with the study. The second attention check item asked: *Is the following statement true:* 'I have been to every country in the world.'? *Please answer honestly and to the best of your knowledge: Definitely untrue, Untrue, True, Definitely true.* Participants who chose either 'True' *or* 'Definitely true' were considered inattentive and were prevented from completing the study. Table 1 summarises the demographic characteristics of the final sample who successfully completed the study.

### Design
A 3 × 6 between-subjects experimental study (for wine and beer) in which participants were randomised to one of 18 groups with one of three levels of verbal *descriptor (Low* vs. *Super Low* vs. *No verbal descriptor)* and six levels of %ABV (five levels varying for wine and beer, and no level given).

### Labels
Based on existing evidence, two low alcohol verbal descriptors (*Low* and *Super Low*) were selected for the purposes of this study.[18] In the present research, we examined the impact of adding %ABV to these two verbal descriptors. We tested a range of %ABV (wine: 0%, 4%, 6%, 8%, 10%, and beer: 0%, 1%, 2%, 3%, 4%). As a control, we had products labelled with no %ABV. For the analyses, we considered that a label with *No verbal descriptor* and No %ABV corresponds to an average (regular) strength product, that is, 12.9% wine and 4.2% beer. The recent study by Vasiljevic and colleagues reported that weekly wine and beer drinkers were able to correctly gauge the %ABV of wines and beers denoted as regular strength.[18] We therefore reasoned that if participants are presented with a product labelled without verbal or numerical information on strength (as they currently appear for sale), they will assume that the product denotes a regular (average) strength wine/beer available on the market. For the labels combining *Low* or *Super Low* verbal descriptors with No %ABV, we used average perceptions of strength that were obtained in Vasiljevic *et al.*[18] wine (*Low* 6.7%, *Super Low* 3.5%), beer (*Low* 2.7%, *Super Low* 1.3%). Two sample labels (one in wine and one in beer) are shown in figure 1.

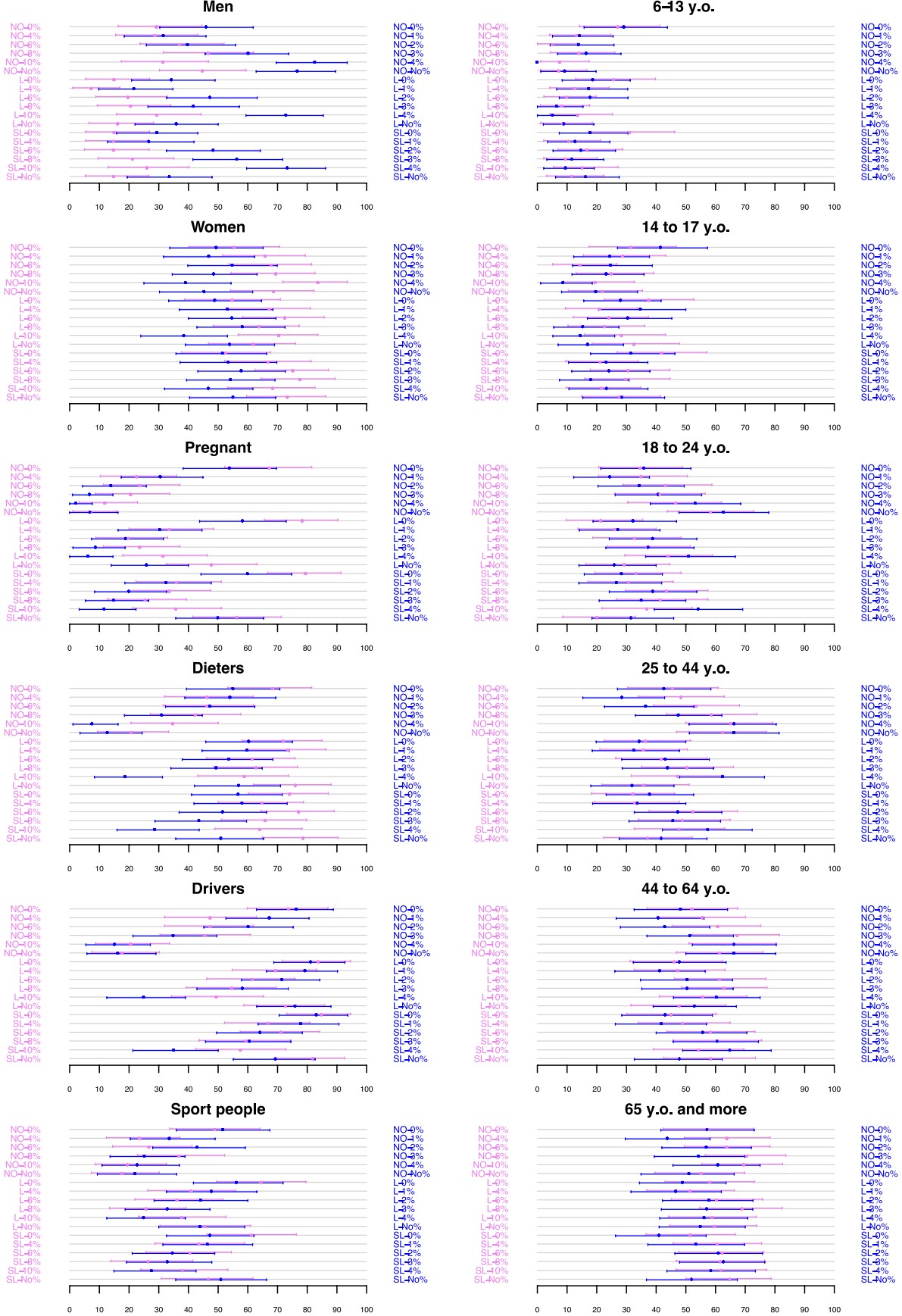

**Figure 2** Proportion of participants (x-axis) considering each drink (Wine in violet, Beer in blue) as likely to appeal for consumption by the different target groups as a function of verbal descriptor and %ABV (y-axis). Arrows correspond to confidence intervals with a global type I error of 5% per target group (Dunn–Šidák multiplicity correction).

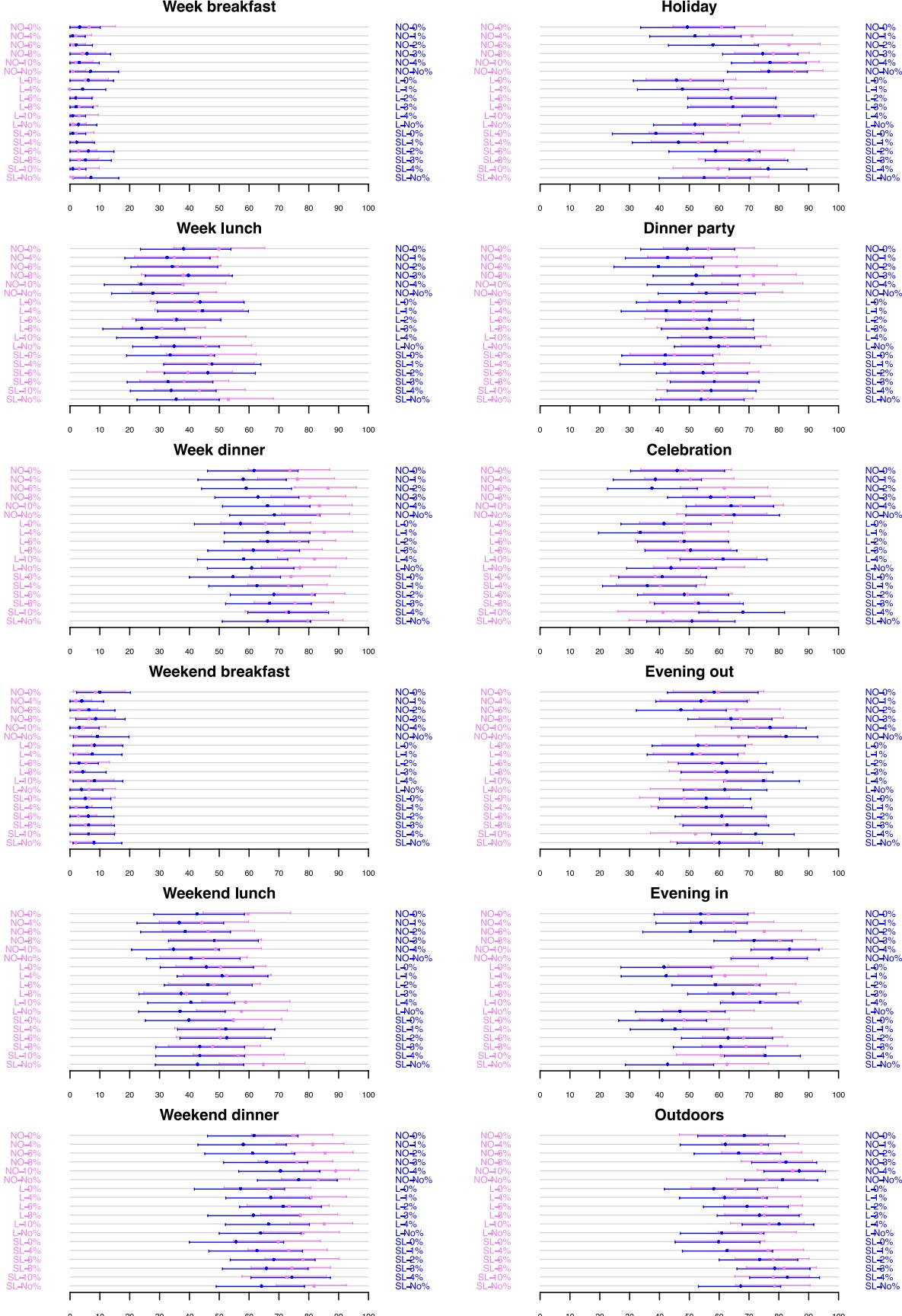

**Figure 3** Proportion of participants (x-axis) considering each drink (Wine in violet, Beer in blue) as likely to appeal for consumption on different target occasions as a function of verbal descriptor and %ABV (y-axis). Arrows correspond to confidence intervals with a global type I error of 5% per target occasion (Dunn–Šidák multiplicity correction).

**Table 2** PCA estimates on perceived target groups in wine and beer drinkers.

| | Wine | | | | Beer | | | |
|---|---|---|---|---|---|---|---|---|
| | **1st Component** | | **2nd Component** | | **1st Component** | | **2nd Component** | |
| **Target groups** | **Est** | **Sig** | **Est** | **Sig** | **Est** | **Sig** | **Est** | **Sig** |
| Men | −0.218 | * | **−0.428** | * | **0.422** | * | −0.268 | * |
| Women | −0.135 | * | 0.298 | | −0.086 | * | 0.178 | |
| Pregnant | **0.528** | * | **−0.511** | * | **−0.370** | * | **−0.770** | * |
| Dieters | **0.384** | * | 0.452 | * | **−0.389** | * | 0.192 | |
| Drivers | **0.497** | * | 0.220 | * | **−0.532** | * | 0.086 | |
| Sportspeople | **0.325** | * | −0.201 | | −0.237 | * | −0.183 | |
| 6–13 | 0.137 | * | **−0.285** | * | −0.105 | * | −0.181 | * |
| 14–17 | 0.128 | * | −0.167 | | −0.128 | * | −0.240 | * |
| 18–24 | −0.202 | * | −0.180 | | 0.237 | * | −0.223 | * |
| 25–44 | −0.231 | * | −0.157 | | 0.257 | * | −0.248 | * |
| 44–64 | −0.147 | * | 0.051 | | 0.179 | * | −0.111 | |
| 65+ | −0.068 | * | 0.074 | | 0.070 | * | 0.107 | |

Note: Target groups significantly contributing to a PCA dimension are denoted with * in column Sig. Large contributions are shown in bold. In Wine, the 1st PCA component explained 78.433% of the variance, and the 2nd component 11.046% of the variance. In Beer, the 1st PCA component explained 83.355% of the variance, and the 2nd component 8.759% of the variance.

## Measures

### Primary outcome measures

The first item assessed perceptions of the target groups of the differently labelled beverages: 'This wine/beer is likely to appeal to: Men, Women, Pregnant Women, Dieters, Drivers, Sportspeople, Those aged 6–13 years, Those aged 14–17 years, Those aged 18–24 years, Those aged 25–44 years, Those aged 45–64 years, Those aged 65 and older'.

The second item measured perceptions of the target occasions for consumption of the differently labelled beverages: 'This wine/beer is likely to be consumed during these occasions: Weekday breakfast, Weekday mid-day meal, Weekday evening meal, Weekend breakfast, Weekend mid-day meal, Weekend evening meal, Holiday, Dinner party, Celebration, Evening out, Evening at home, Outdoors barbecue/picnic'.

Responses on both items were recorded as: *Definitely not, Probably not, Probably yes, Definitely yes*. For analyses participants' responses were dichotomised as our interest lies in the proportion of participant who perceive drinks labelled with different verbal descriptor x %ABV combinations as likely to appeal to different target groups and target occasions. Sensitivity analyses considering the four-level Likert scale responses instead of the dichotomised responses yielded similar results.

### Individual difference measure: risky drinking

Level of risky drinking was assessed using the AUDIT-C,[19] the first three items of the Alcohol Use Disorders Identification Test (AUDIT).[20] A sample item asked 'How many drinks containing alcohol do you have on a typical day when you are drinking?' responses ranged from *1 or 2, 3 or 4, 5 or 6, 7 to 9, 10 or more*. Following recommendations, responses to the AUDIT-C were summed, and dichotomised to denote riskier (scoring equal to or greater than 5) versus less risky drinking patterns (scoring below 5).[21]

### Demographic characteristics

The following were recorded: age, gender, ethnicity and socio-economic status (assessed using individual-level measures of highest educational qualification, income and occupational status, and neighbourhood-level deprivation assessed via the postcode-based Index of Multiple Deprivation).[22]

## Procedure

The protocol was approved by the University of Cambridge Psychology Research Ethics Committee (Pre.2015.077). The study was carried out in the period from June to August 2016. Participants were sampled from an existing market research agency panel. The existing panel was prescreened and only those who reported drinking alcohol at least once a week were eligible to participate in the study. After stating their preferred drink, participants were randomised by the survey software platform (*Qualtrics*) to see one of the 18 alcohol labels placed either on a bottle of wine or beer. Participants who expressed equal liking for wine and beer were randomly assigned to either the wine or beer surveys. Participants completed the study outcome measures while viewing the assigned product.

## Patient and public involvement

Patients were not involved in the planning of this study, since the study was not aimed at patient samples. Research members of the Behaviour and Health Research Unit, University of Cambridge provided comments on the questionnaire materials prior to testing. The research

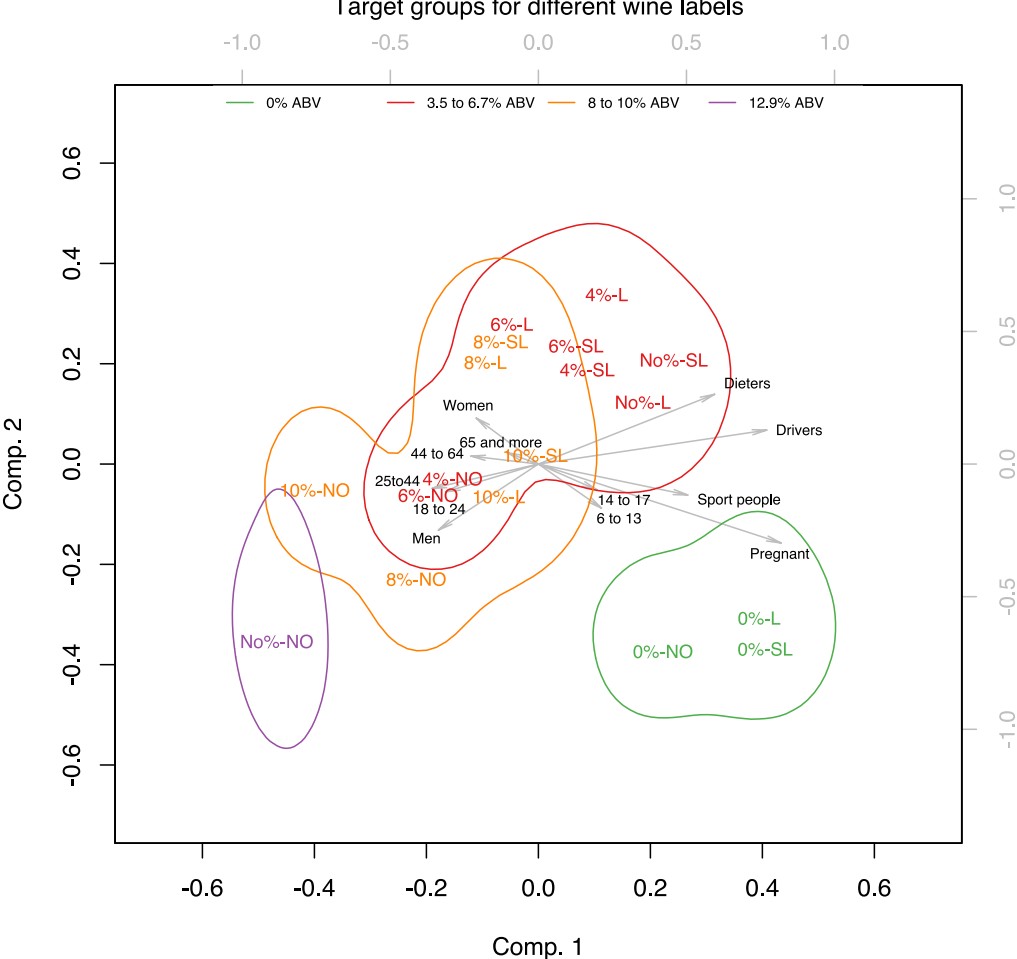

**Figure 4** Biplot of the PCA for target groups in Wine, showing the score of each drink with a given verbal descriptor and %ABV combination as well as the loadings of each target group on the two first principal components. The verbal descriptor and %ABV combinations were categorised into four distinct colour-coded groups corresponding to different bands of %ABV [Wine: 0%ABV, 3.5%–6.7%ABV, 8%–10%ABV, 12.9%ABV]. The coloured contours correspond to 95% confidence areas of these four bands.

members gave suggestions on how the materials could be edited to make them easier to understand for participants sampled from the general UK population. The research members who piloted the materials were not involved in study recruitment and conduct. Participating research members were informed of the study results via a short presentation during one of the regular meetings of the group.

### Analysis

We first conducted descriptive analyses on the raw data to examine the proportion of participants who considered each labelled beverage (as a function of the verbal descriptor x %ABV experimental combinations) as likely to appeal to the different target groups and target occasions. We then performed principal component analyses (PCAs) to reduce the dimensions of target groups and target occasions for each of the 18 products for wine and beer. PCA allows reduction of the multiple target groups and occasions by exploiting the correlation structure

observed in the data.[23] We reduced the dimension of the different target groups and occasions, and projected the verbal descriptors x %ABV experimental conditions across the two first dimensions of the principal components (defined as linear combinations of the target groups and occasions). We used non-parametric bootstraps to define 95% CIs for the loadings and 95% confidence areas for the projections of verbal descriptor x %ABV on the PCA scatterplot.

### RESULTS

Figures 2 and 3 are graphical representations of the proportion of participants who considered each beverage as likely to appeal to the different target groups and target occasions as a function of the verbal descriptor and %ABV experimental combinations (on the y-axis). Arrows correspond to 95% CIs with a global type I error of 5% per target group and target occasion using a Dunn–Šidák

## Target groups for different beer labels

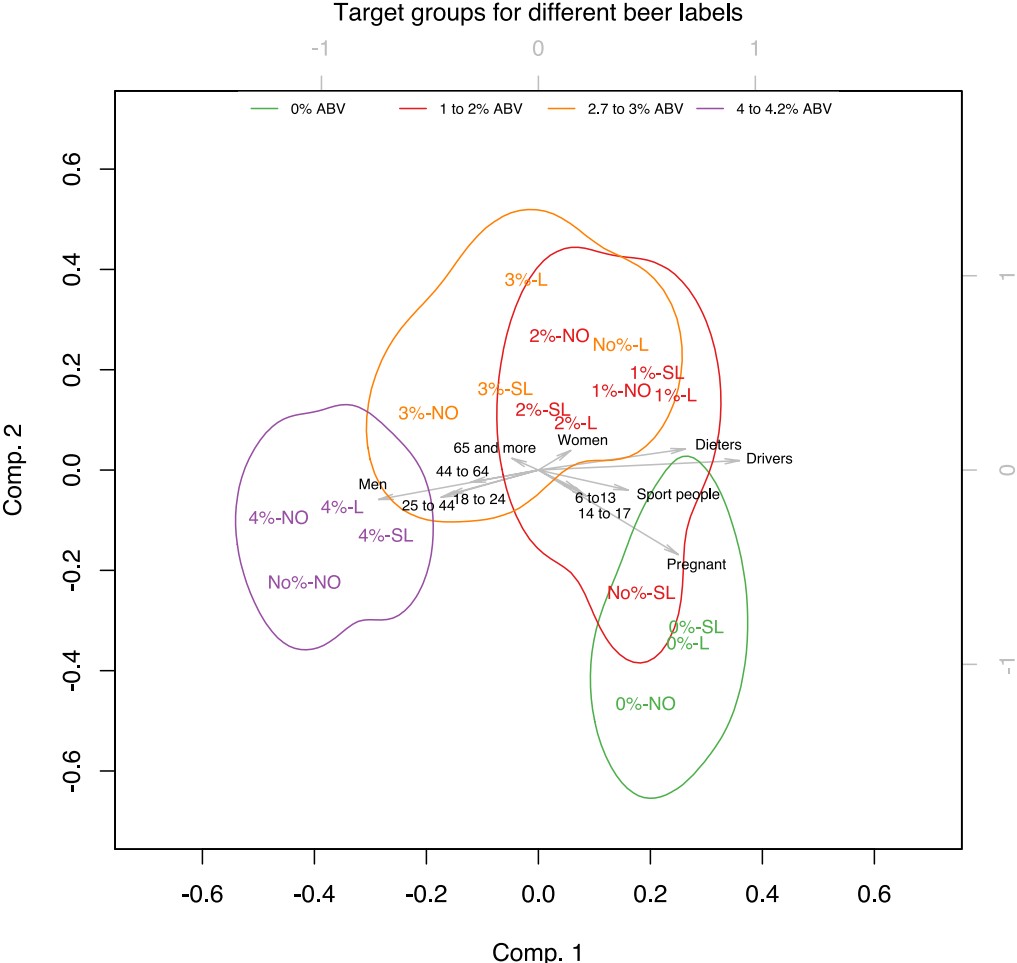

**Figure 5** Biplot of the PCA for target groups in Beer, showing the score of each drink with a given verbal descriptor and %ABV combination as well as the loadings of each target group on the two first principal components. The verbal descriptor and %ABV combinations were categorised into four distinct colour-coded groups corresponding to different bands of %ABV [Beer: 0%ABV, 1%–2%ABV, 2.7%–3ABV, 4%–4.2%ABV]. The coloured contours correspond to 95% confidence areas of these four bands.

multiplicity correction. As can be seen in figure 2, a greater proportion of participants considered pregnant women, dieters, drivers, and sportspeople as targets of drinks labelled with verbal and/or numerical descriptors of lower alcohol strength. By contrast, men, women, and those aged 18–24, 25–44, and 45–64 years were considered as targets of drinks labelled as higher in strength. Those aged 65+ years were considered as targets for both lower and higher alcohol strength wines and beers.

Figure 3 shows that a higher proportion of participants considered weekday lunches as target occasions for drinking lower strength wines and beers. On the other hand, weekend lunches were considered as likely targets for the consumption of both lower and higher strength wines and beers. Weekday and weekend dinners, dinner parties, holidays, celebrations, outdoor events, evenings in and out were considered as target occasions for the consumption of higher strength wines and beers by a greater proportion of participants. Very few participants

considered weekday and weekend breakfast as target occasions for consuming wine and beer, irrespective of whether it was labelled as lower or higher in strength.

We performed PCAs to reduce the dimensions of the target groups and occasions and analysed the results of the first two principal components as they explained a large proportion of the variance of the original dimensions [Target groups: $89.48\%_{wine}$, $92.11\%_{beer}$; Target occasions: $83.52\%_{wine}$, $87.35\%_{beer}$].

### Target groups

In the PCA defining the perceived target groups for each of the different wine and beer products, the first component was mainly defined by dieters, drivers, pregnant women and sportspeople. The second component was mainly defined by pregnant women for beer, while for wine, it was defined by pregnant women, those aged 6 to 13 in one quadrant, and men (see table 2).

**Table 3** PCA estimates on perceived target occasions in wine and beer drinkers.

| | Wine | | | | Beer | | | |
| --- | --- | --- | --- | --- | --- | --- | --- | --- |
| | 1st Component | | 2nd Component | | 1st Component | | 2nd Component | |
| Target occasions | Est | Sig | Est | Sig | Est | Sig | Est | Sig |
| Week breakfast | −0.016 | | 0.019 | | −0.006 | | −0.124 | |
| Week lunch | −0.171 | | **−0.556** | * | 0.164 | * | **−0.619** | * |
| Week dinner | 0.193 | * | −0.256 | | −0.103 | * | **−0.349** | * |
| Weekend breakfast | −0.031 | | −0.083 | | −0.003 | | −0.088 | |
| Weekend lunch | −0.121 | | **−0.674** | * | 0.054 | | **−0.598** | * |
| Weekend dinner | 0.227 | * | −0.233 | | −0.170 | * | **−0.300** | * |
| Holiday | **0.535** | * | −0.099 | | **−0.488** | * | −0.050 | |
| Dinner party | 0.343 | * | −0.237 | | −0.177 | * | −0.086 | |
| Celebration | 0.375 | * | −0.069 | | **−0.392** | * | −0.002 | |
| Evening out | 0.288 | * | −0.025 | | **−0.336** | * | 0.048 | |
| Evening home | **0.441** | * | 0.194 | | **−0.524** | * | 0.031 | |
| Outdoors | 0.211 | * | 0.022 | | −0.346 | * | −0.106 | |

*Note.* Target occasions significantly contributing to a PCA dimension are denoted with * in column Sig. Large contributions are shown in bold. In Wine the 1st PCA component explained 71.182% of the variance, and the 2nd component 12.340% of the variance. In Beer the 1st PCA component explained 78.783% of the variance, and the 2nd component 8.571% of the variance.

Figures 4 and 5 show the projection of the verbal descriptor x %ABV experimental conditions on the two first principal components defined as above. The verbal descriptor x %ABV experimental conditions were categorised into four distinct colour-coded groups corresponding to different bands of %ABV [Wine: 0%ABV, 3.5%–6.7%ABV, 8%–10%ABV, 12.9%ABV; Beer: 0%ABV, 1%–2%ABV, 2.7%–3ABV, 4%–4.2%ABV]. The coloured contours correspond to 95% confidence areas of these four bands. The chosen colour coding demonstrates that participants were guided more by the %ABV rather than the verbal descriptor on the label since experimental conditions denoting adjacent levels of %ABV appear closer while labels sharing the same verbal descriptor were located at distant positions. The confidence areas of the projections of the verbal descriptor x %ABV experimental conditions on the two main dimensions of the PCA show the same pattern for both wine and beer. The biplots demonstrate that participants believed pregnant women, sportspeople and those aged 6–13 and 14–17 years old were the target groups for products containing 0%ABV or products containing the verbal descriptors *Low* or *Super Low* in their labels, whereas the other target groups (made up of men, women, and those aged above 18) were the target groups for products containing 8%–10%ABV, 12.9%ABV in their label for wine and 2.7%–3ABV, 4%–4.2%ABV in the label for beer.

### Target occasions

In the PCA defining the occasions targeted for each of the different wine and beer products, the first component was mainly defined by holidays, dinner parties, celebrations, and evenings at home (with an additional occasion of evenings out for beer). The second component in beer was mainly defined by weekday lunch, weekend lunch, weekday dinner and weekend dinner, while in wine, it was defined by weekday lunch and weekend lunch (see table 3).

Figures 6 and 7 show the projection of the verbal descriptor x %ABV experimental conditions on the first two principal components defined as above. As for the target groups, the verbal descriptor x %ABV experimental conditions were categorised into four distinct colour-coded groups corresponding to different bands of %ABV. The confidence areas of these four bands of %ABV show the same pattern on the biplots which show the projections of the first two dimensions. The biplots demonstrate that participants rated weekday lunches as the target occasions for consuming products containing 0%ABV or the verbal descriptors *Low* or *Super Low* in their labels. Weekend lunches were the perceived target occasions for consuming both lower and higher strength wines and beers.

Products labelled with 8%–10%ABV, 12.9%ABV in wine and 2.7%–3ABV, 4%–4.2%ABV in beer were rated as targeting dinner/evening occasions, including parties, holidays and celebrations. As before the chosen colour coding demonstrates that participants were guided more by the %ABV rather than the verbal descriptor on the label since experimental conditions denoting adjacent levels of %ABV appear closer while labels sharing the same verbal descriptor were located at distant positions.

### DISCUSSION

Analyses of the perceived target groups for consumption showed that participants perceived 0% and other lower

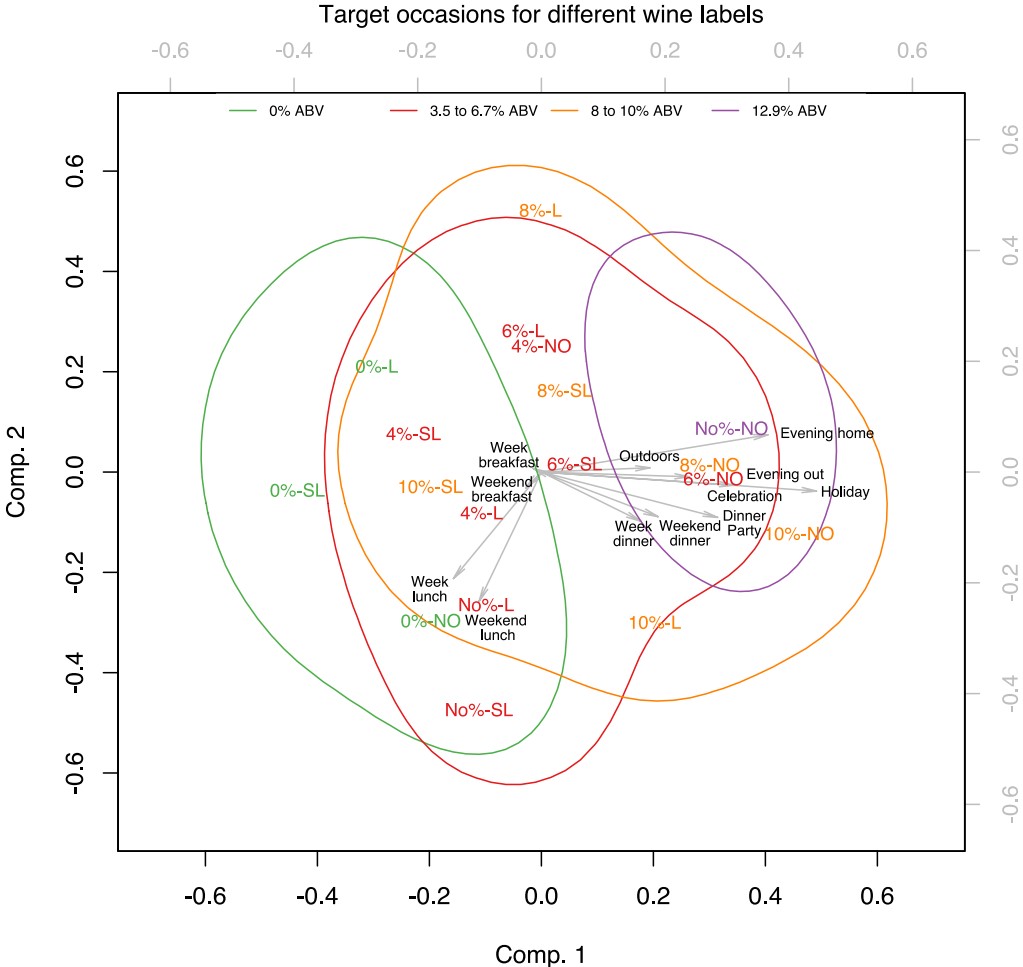

**Figure 6** Biplot of the PCA for target occasions in Wine, showing the score of each drink with a given verbal descriptor and %ABV combination as well as the loadings of each target occasion on the two first principal components. The verbal descriptor and %ABV combinations were categorised into four distinct colour-coded groups corresponding to different bands of %ABV [Wine: 0%ABV, 3.5%–6.7%ABV, 8%–10%ABV, 12.9%ABV]. The coloured contours correspond to 95% confidence areas of these four bands.

strength labelled drinks as targeting pregnant women, dieters, drivers, sportspeople, and those under age. In terms of targeted occasions, 0% and other lower strength drinks were perceived as targeting weekday lunchtimes. Products containing higher levels of %ABV were rated as targeting men, women, and those aged above 18, as well as dinner/evening occasions, including parties, holidays, and celebrations. Weekend lunches were considered as the target occasions for both lower and higher strength wines and beers. These findings suggest that the general population of weekly drinkers perceives lower strength wines and beers as an extension to regular strength alcohols, rather than as a substitute product, which may have unintended consequences for overall consumption levels.[12–15]

It is important to note that extending the target groups and occasions of alcohols labelled as 0%ABV would be beneficial to public health, since these products do not contain any alcohol. Of concern is that weekly wine and beer drinkers also perceived products labelled with numerical indicators of higher %ABV, and the verbal descriptors *Low* and *Super Low* as likely to appeal for consumption to more groups, and on more occasions that would traditionally have been reserved for consuming soft drinks and no-alcohol alternatives.[see also[17]].

These results are compatible with the possibility that lower strength alcohol labelling may paradoxically serve to increase total alcohol consumption by positioning lower strength products as suitable for consumption by target groups that may currently consume soft drinks and no-alcohol alternatives (such as drivers and dieters).[12 13] Any changes to current labelling legislation that extend the range of lower alcohol content labelling may thus result in increasing rather than decreasing total levels of alcohol consumed [see also[16]]. However, the certainty attached to this is very low given the nature of this evidence. Careful monitoring of marketing and sales data will be required to examine if imminent changes

Target occasions for different beer labels

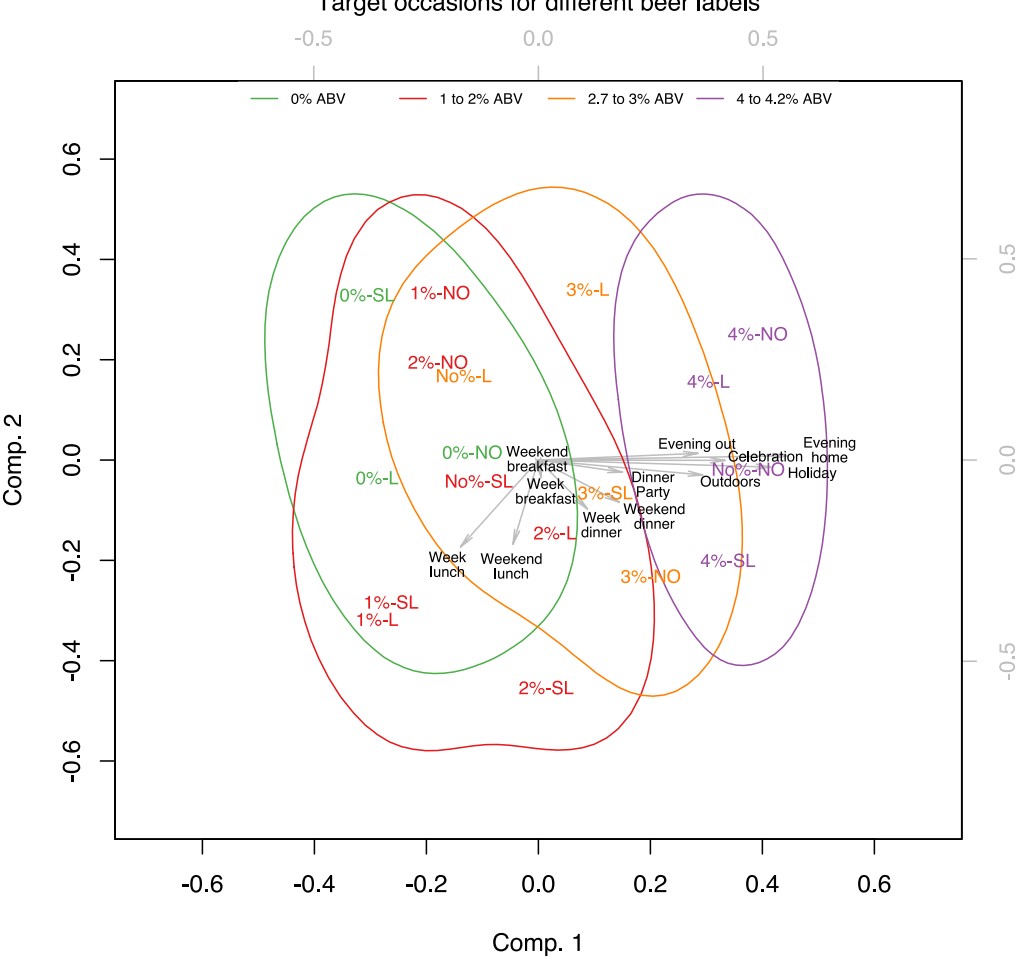

**Figure 7** Biplot of the PCA for target occasions in Beer, showing the score of each drink with a given verbal descriptor and %ABV combination as well as the loadings of each target occasion on the two first principal components. The verbal descriptor and %ABV combinations were categorised into four distinct colour-coded groups corresponding to different bands of %ABV [Beer: 0%ABV, 1%–2%ABV, 2.7%–3ABV, 4%–4.2%ABV]. The coloured contours correspond to 95% confidence areas of these four bands.

to lower strength alcohol labelling leads to unintended consequences.

Our results are compatible with recent analyses of the marketing messages on producers' and retailers' websites for low/er and regular strength wines and beers sold online by the four main supermarkets in the UK. These analyses showed that low/er strength products were marketed not as substitutes for higher strength products but as ones that can be consumed on additional occasions with an added implication of healthiness.[17] One possibility is that the online marketing for lower strength products has shaped perceptions of the general population regarding the wider range of groups and occasions targeted by lower strength alcohol products. A direct test of this hypothesis is, however, beyond the current study. Future studies could usefully extend the present findings by testing whether there is a direct link between producers' and retailers' marketing messages,

and consumers' perceptions regarding the target groups and occasions for consumption of lower content alcohols.

The present data also show that consumers are guided more by the numerical information of %ABV rather than the verbal descriptor of strength, since the perceived target groups and occasions clustered among products denoted with adjacent %ABV, rather than adjacent verbal descriptors of strength. This suggests that alcohol strength may be better communicated by numerical rather than verbal information. The empirical and theoretical implications of this finding merit further testing.

### Strengths and limitations with suggestions for future research

Strengths of the present research include the experimental design and the large sample of weekly drinkers sampled from a nationally representative panel of the UK population. Whether these findings can be generalised to other cultural contexts cannot be inferred from the

present study. Replications in other countries are needed to assess this.

The use of fictitious non-branded labels further strengthens the conclusions that can be drawn from the present study, since we could rule out the confounding influence of brand recognition and loyalty. Nevertheless, future research should extend the present findings by examining how participants respond to existing branded labels, since this may affect participants' perceptions and ultimately their selection of alcohol products for consumption.

The present research is limited in using an online sampling frame with no behavioural outcome measure. Whether the self-reported perceptions of the target groups and occasions of lower strength alcohol labelling reflect actual drink choices of consumers in the real world is not certain. Future research could examine drink selection in naturalistic contexts, to provide behavioural indicators of the type of consumer and type of occasion that are associated with the choice to consume lower strength alcohols.

## Policy implications

Our study examined the perceived target groups and occasions of lower strength alcohol labelling with the view of aiding decision-making in the context of imminent legislative changes to alcohol labelling rules in the UK.[4] The present findings suggest that extending the range of verbal descriptors and the numerical threshold (above the current 1.2%ABV) for labelling of lower strength alcohols may carry unintended consequences, such as increasing the occasions when alcohol is consumed, and increasing the consumer base to groups that have traditionally been associated with the consumption of soft drinks or other no-alcohol drinks. These findings fit recent findings that online marketing of lower strength alcohols was more likely to suggest extended occasions suitable for alcohol consumption, such as drinking during lunchtimes and during sports-activities.[17] Careful monitoring of sales data will be needed to ascertain if these marketing messages and consumers' perceptions translate into actual increases in overall alcohol consumption. Taken together these findings suggest that labels not highlighting the lower alcohol strength of drinks may be more effective in reducing overall alcohol consumption than those in which the lower alcohol content is highlighted.[see also [24]]

## CONCLUSIONS

Weekly wine and beer drinkers perceived that products labelled as lower in strength were targeted at non-traditional occasions (such as weekday lunchtimes) and groups (such as pregnant women, dieters, drivers) thereby suggesting that alcohol can be consumed more often and by more people. Further research is needed to test whether these perceptions translate into actual behaviour.

**Acknowledgements**  We would like to thank Mario Weick for help with the piloting of the study, preparation of study materials, as well as for commenting on an earlier version of this manuscript. We would also like to thank the members of the Behaviour and Health Research Unit for their comments and suggestions on the study materials.

**Contributors**  All authors collaborated in designing the study and revising the manuscript critically for important intellectual content. D-LC carried out the sample size calculations, and data analyses. MV wrote the initial draft of the manuscript, D-LC and TMM provided critical revisions to the manuscript. All authors read and approved the final version of the manuscript. All authors agree to be accountable for all aspects of the work in ensuring that questions related to the accuracy or integrity of any part of the work are appropriately investigated and resolved.

**Funding**  This report is independent research commissioned and funded by the National Institute for Health Research Policy Research Programme (Policy Research Unit in Behaviour and Health [PR-UN-0409-10109]) and a National Institute for Health Research Response Mode Grant [PR-ST-0615-10012]). The views expressed in this publication are those of the authors and not necessarily those of the NHS, the National Institute for Health Research, the Department of Health and Social Care or its arm's length bodies, and other Government Departments. The final version of the report and ultimate decision to submit for publication was determined by the authors.

**Competing interests**  None declared.

**Patient consent for publication**  Not required.

**Provenance and peer review**  Not commissioned; externally peer reviewed.

**Data sharing statement**  Data are available from the authors upon request.

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
