## [Reviewer comments · BMJ Open]

ARTICLE DETAILS

TITLE (PROVISIONAL)	What are the perceived target groups and occasions for wines and beers labelled with verbal and numerical descriptors of lower alcohol strength? An experimental study
AUTHORS	Vasiljevic, Milica; Couturier, Dominique-Laurent; Marteau, Theresa

VERSION 1 - REVIEW

REVIEWER	Dr. Montana Halliday University of British Columbia Faculty of Medicine
REVIEW RETURNED	14-Jun-2018

GENERAL COMMENTS	Page 5, Line 37: "However, this study only focused on the content of the marketing messages, and did not examine how consumers perceive and respond to alcohol products labelled as lower in strength." I think this could be clarified a bit because it sounds like this study actually does look at how consumers perceive alcohol products labelled as lower in strength. Maybe just cut that part out and leave "respond to.." Page 6, Line 27: I think this is actually the main meaningful difference in your study compared to the other one cited. Your study doesn't look at how appealing the drinks are, but who consumers think they are suitable for. These findings may provide insight into why the other study found lower strengths less appealing. Page 7, Line 35: Really? In the UK beverages are not labelled with their strengths? I thought they were labelled in units. I think in general this is a good paper. However, there is a limitation that all of the tested strengths are lower than regular or high strength. I know there is the assumption that the participants thought the unlabelled beverage was regular strength, but that can't be proven. How can an accurate comparison be made if only low strengths are being surveyed? I think it would have been better to include a "regular strength" and "higher strength" beverage for comparison.
--

REVIEWER	Chelsie Young Rowan University United States
REVIEW RETURNED	31-Aug-2018

GENERAL COMMENTS	This manuscript experimentally examined a large sample of UK individuals' perceptions of which categories of people and on which occasions people might be most likely to consume a lower strength alcohol. Alcoholic beverage labels for wine or beer depending on the participants' preference were manipulated to contained either %ABV and verbal descriptors of alcohol content or just the verbal descriptors to understand which label is more informative. This study is an interesting step into an important area. The sample size is large and the results can have notable policy implications. However, there are some concerns in the level of detail reported, particularly in the Method section, and the conclusions drawn from the data. Major Points: The authors don't provide specific hypotheses in the introduction. What did they expect to find? More detail is needed here. The authors state that including the %ABV would make the labels more understandable and differentiate more clearly the regular strength beverages from the low strength beverages, but do not overtly state their hypotheses for how this will play out in their results. If there were no a priori hypotheses, this could be made more clear by framing the last paragraph of the introduction around this study as an exploration of perceptions of low strength alcoholic beverages. Brief summaries of the overall demographic information for the sample (e.g., age, gender, race, ethnicity, and SES) could be added to the participants section of the method to give readers an idea of who participants were (not divided up by wine or beer drinkers). More detail is needed regarding the process by which inattentive responders were screened out of the study. What were the attention checks mentioned on page 6 in the participants subsection of the method? How many people failed attention checks? Page 6 in the participants subsection mentions a pilot study but does not provide what was examined in the pilot study. More information would be helpful in what was included in the pilot study and how that informed the present study. In the method section on page 8 the authors state that the results were dichotomized. Did the authors test whether the results are the same when the data are treated as continuous compared to dichotomous? The degree to which people agree or disagree might be more informative. It is somewhat unclear exactly what the study consisted of. Did participants only see 1 alcohol label and were asked for that beverage who might drink it and on what sorts of occasions? Was there a direct test of the labels containing both verbal descriptors and ABV% versus just including verbal descriptors? Was participants' own drinking assessed to test whether own drinking was associated with their perceptions? Were participants' descriptive or injunctive norms assessed, in particular, what occasions they think others drink or that they find acceptable to drink any alcohol, regardless of strength? Did perceptions differ as a function of participant age or gender? The authors seem to argue that offering a low strength alcohol would increase overall alcohol consumption because they found
--

	that participants reported that the substance might appeal to a wider variety of people and that others might drink this lower strength alcohol in more situations than regular strength alcohol use. However, the results do not suggest that more categories of people will drink this beverage and on more occasions than regular strength alcohol. The lack of behavioral outcome data makes this conclusion an extrapolation that is not necessarily support by data in this study. There is no evidence from this study that people would drink more of the lower strength alcohol or that by potentially increasing drinking occasions, that problems would result, so these potentialities should be walked back some throughout the paper. Throughout the study the authors use language that participants were rating acceptability, suitability, or approval of drinking the lower strength alcohol by specific subgroups of people and on particular occasions. However, the participants were asked how likely those people would drink lower strength alcohol and what occasions people would likely consume the lower strength alcohol. Likelihood is not the same as approval or suitability. The participants were not necessarily conveying that they approved of these individuals and occasion for drinking low strength alcohol. That distinction is very important and the paper should be revised to reflect that distinction. For example, the abstract says "...the kind of person who would find the beverage most appealing, and the types of occasions on which the beverage could be drunk", to more accurately reflect the findings, the authors could change this to "subtypes of people they perceived to be more likely to drink the lower strength alcohol and occasions at which people would be more likely to drink it." Or something to that effect. Minor Points: Page 8 line 45 and 46 appear larger than the text in the rest of the paragraph. Page 9 line 52 insert and between drivers and sportspeople Page 13 line 3 underage should be one word The font in the tables is not times new roman On page 6 line 39, APA style does not allow for numbers at the beginning of sentences. Add a comma after the work link on page 6 line 43.
--	---

VERSION 1 – AUTHOR RESPONSE

Reviewer(s)' Comments to Author:

Reviewer: 1

Reviewer Name: Dr. Montana Halliday

Institution and Country: University of British Columbia Faculty of Medicine

Please state any competing interests or state 'None declared': None to declare

Page 5, Line 37: "However, this study only focused on the content of the marketing messages, and did not examine how consumers perceive and respond to alcohol products labelled as lower in strength." I think this could be clarified a bit because it sounds like this study actually does look at how consumers perceive alcohol products labelled as lower in strength. Maybe just cut that part out and leave "respond to.."

- We believe there may be some confusion as to which study we are referring to on page 5, line 37. The study we are discussing in those lines (Vasiljevic et al., 2018) examined the marketing messages on retailers' and producers' websites and did not examine

consumers' perceptions and responses. As discussed on page 5, line 37, in Vasiljevic et al. (2018), only the actual marketing materials were coded. For more details please see: Vasiljevic M, Coulter L, Petticrew M, Marteau TM. Marketing messages accompanying online selling of low/er and regular strength wine and beer products in the UK: a content analysis. *BMC Public Health* 2018;18:147. doi:10.1186/s12889-018-5040-6

- We also provide below the relevant excerpt from the revised manuscript to show how this sentence fits with the rest of the paragraph describing this study:

<< Another recent study analysed the content of marketing messages on producers' and retailers' websites for lower and regular strength wines and beers sold online by the four main supermarkets in the UK.[17] This study found that lower strength alcohols were marketed not as substitutes for higher strength products but as ones that can be consumed on additional occasions.[17] Furthermore, lower strength wines and beers were more often marketed with claims to health benefits. These findings raise a broader question of the extent to which lower strength alcohol products will contribute to a public health strategy to reduce alcohol consumption and associated harms. However, this study only focused on the content of the marketing messages, and did not examine how consumers perceive and respond to alcohol products labelled as lower in strength. >>

Page 6, Line 27: I think this is actually the main meaningful difference in your study compared to the other one cited. Your study doesn't look at how appealing the drinks are, but who consumers think they are suitable for. These findings may provide insight into why the other study found lower strengths less appealing.

- The prior study we are referring to on page 6, line 27 only assessed participants' perceptions of strength and appeal of different verbal descriptors of lower alcohol strength. The study was also limited by the fact that the verbal descriptors of strength were not coupled with numerical information of %ABV on actual labels. For more information see: Vasiljevic M, Couturier DL, Marteau TM. Impact of low alcohol verbal descriptors on perceived strength: An experimental study. *Br J Health Psychol* 2018;23:38-67. doi:10.1111/bjhp.12273
- As described on page 6 we discuss how our present study under review fills the gaps in the extant literature. See page 6, lines 12-31 of the revised manuscript; and the extract below for your ease of reference:

<< The study was limited because the verbal descriptors of strength were not coupled with numerical information of percentage alcohol by volume (%ABV) on actual labels. Furthermore, the study only assessed participants' perceptions of strength and appeal of the different verbal descriptors. The current study aimed to fill this gap by combining numerical indicators of %ABV with a selection of verbal descriptors identified as most differentiating and understandable in the study by Vasiljevic and colleagues.[18] For the current study we developed purposefully tailored labels so that we could control for any extraneous influences of participants' prior brand preference. The current study also extended previous studies by examining what weekly drinkers of wine and beer perceived to be the target groups and occasions of drinks labelled as lower in alcohol strength. >>

- We believe this describes in sufficient detail the contrast between our previous study - that tested perceptions of strength and appeal of different verbal descriptors of lower alcohol strength - and our current study - that tested the perceived target groups and occasions for consuming drinks labelled with different (verbal and/or numerical) indicators of alcohol strength.

Page 7, Line 35: Really? In the UK beverages are not labelled with their strengths? I thought they were labelled in units.

- Information on alcohol content (%ABV) and unit content usually appears on the back and not on the front label of wines and beers currently sold in the UK. Please see some image examples of popular wines and beers currently sold in the UK provided below. The front-of-bottle labels from these products provided the basis for our own tailored labels used in the current study.

- Our study only examined the impact of front-of-bottle manipulated labels.
- We provide below an image of the Regular label for wine.

I think in general this is a good paper. However, there is a limitation that all of the tested strengths are lower than regular or high strength. I know there is the assumption that the participants thought the unlabelled beverage was regular strength, but that can't be proven. How can an accurate comparison

be made if only low strengths are being surveyed? I think it would have been better to include a "regular strength" and "higher strength" beverage for comparison.

- We thank you for your positive assessment of our paper. Here we would like to further clarify our reasoning pertaining to the different labels used in the study.
- Our Regular label served as a control group label and, as such, was similar to the other labels other than not containing any verbal or numerical indicators of strength (see also our response to your previous comment). In other words, this was a label on a drink that does not call attention to its alcohol content (either verbally or numerically). The recent study by Vasiljevic, Couturier, and Marteau (2018) [Reference number 18 in the revised manuscript] reported that weekly wine and beer drinkers were able to correctly gauge the %ABV of wines and beers denoted as regular strength. For further information see: Vasiljevic M, Couturier DL, Marteau TM. Impact of low alcohol verbal descriptors on perceived strength: An experimental study. *Br J Health Psychol* 2018;23:38-67. doi:10.1111/bjhp.12273
- We therefore reasoned that, if participants are presented with a product labelled without %ABV information, they will assume that the product denotes a regular (average) strength wine/beer available on the market. Our findings support this hypothesis, as can be seen in Figures 4 and 5 graphically presenting the results of the PCA analyses. The drinks with labels which contained no verbal or numerical descriptors of strength clustered with the other drinks labelled as denoting higher (average alcohol strength [%ABV] on the market i.e. the labels denoted with 4%ABV in beer and 8%ABV or 10%ABV in wine). We are therefore confident that our assumption holds.

Reviewer: 2

Reviewer Name: Chelsie Young

Institution and Country: Rowan University, United States

Please state any competing interests or state 'None declared': None declared

This manuscript experimentally examined a large sample of UK individuals' perceptions of which categories of people and on which occasions people might be most likely to consume a lower strength alcohol. Alcoholic beverage labels for wine or beer depending on the participants' preference were manipulated to contained either %ABV and verbal descriptors of alcohol content or just the verbal descriptors to understand which label is more informative. This study is an interesting step into an important area. The sample size is large and the results can have notable policy implications. However, there are some concerns in the level of detail reported, particularly in the Method section, and the conclusions drawn from the data.

- We appreciate your positive assessment of our work, and hope that we can address all points for improvement that you noted in your review.

Major Points:

The authors don't provide specific hypotheses in the introduction. What did they expect to find? More detail is needed here. The authors state that including the %ABV would make the labels more understandable and differentiate more clearly the regular strength beverages from the low strength beverages, but do not overtly state their hypotheses for how this will play out in their results. If there were no a priori hypotheses, this could be made more clear by framing the last paragraph of the introduction around this study as an exploration of perceptions of low strength alcoholic beverages.

- Since this is the first study of this kind we did not have pre-specified hypotheses. We have clarified this in the last sentence of the Background section (see extract below):

<< Since this is the first type of study to examine drinkers' perceptions of the target groups and occasions of drinks labelled with verbal and numerical information of lower alcohol strength we did not have any pre-specified hypotheses. >>

Brief summaries of the overall demographic information for the sample (e.g., age, gender, race, ethnicity, and SES) could be added to the participants section of the method to give readers an idea of who participants were (not divided up by wine or beer drinkers).

- Many thanks for this suggestion. Due to word count constraints we have opted not to present information on participant demographics twice (both in text and table format). The comprehensive demographic breakdown of all participants who successfully completed the study can be seen in Table 1.

More detail is needed regarding the process by which inattentive responders were screened out of the study. What were the attention checks mentioned on page 6 in the participants subsection of the method? How many people failed attention checks?

- Thanks for prompting us to elaborate on the attention checks we employed in the present study. We have added further information in the Participants subsection of the Methods section explaining the attention checks in more detail. In total, 91 participants failed to complete the beer study, and 72 participants failed to complete the wine study since they failed one of the two attention checks. Please note that Table 1 shows the demographic characteristics of the participants who fully completed the study. The relevant extract we added is also shown below:

<< Attention checks were used to screen-out inattentive responders. Participants were informed in the Information Sheet and Consent Form that there would be attention checks in the online survey, and that failure to complete the attention checks correctly would result in them being prevented from completing the study. Attention was gauged by two items: When was the last time you have flown to Mars? Please answer honestly and to the best of your knowledge: Never/A few days ago/Weeks ago/Months ago. Participants who did not choose the only plausible option of 'Never' were considered inattentive and were prevented from continuing with the study. The second attention check item asked: Is the following statement true: "I have been to every country in the world."? Please answer honestly and to the best of your knowledge: Definitely untrue, Untrue, True, Definitely true. Participants who chose either 'True' or 'Definitely true' were considered inattentive and were prevented from completing the study. Table 1 summarises the demographic characteristics of the final sample who successfully completed the study. >>

Page 6 in the participants subsection mentions a pilot study but does not provide what was examined in the pilot study. More information would be helpful in what was included in the pilot study and how that informed the present study.

- We have now added this additional information. See pages 6-7 and below:

<< Since effect size estimates were not available for the outcomes of interest, the sample size calculations were based on differences in ratings of appeal of different wines and beers labelled with verbal and/or numerical descriptors of lower alcohol strength observed in a pilot study. >>

In the method section on page 8 the authors state that the results were dichotomized. Did the authors test whether the results are the same when the data are treated as continuous compared to dichotomous? The degree to which people agree or disagree might be more informative.

- We did indeed also carry out sensitivity PCA analyses which replicated the results presented in the manuscript. Since the analyses yielded similar results, we chose to present the PCA analyses based on the dichotomised outcomes as our interest lies in the proportion of participant who perceive drinks labelled with different verbal descriptor x %ABV combinations as likely to appeal to different target groups and target occasions. We have added a sentence in the revised manuscript to this effect (see page 9 and below):

<< Sensitivity analyses considering the four-level Likert scale responses instead of the dichotomised responses yielded similar results. >>

It is somewhat unclear exactly what the study consisted of. Did participants only see 1 alcohol label and were asked for that beverage who might drink it and on what sorts of occasions? Was there a direct test of the labels containing both verbal descriptors and ABV% versus just including verbal descriptors?

- Yes, that is correct. The design of the experiment was between-subjects, whereby participants were randomised to see one of the 18 labelled bottles according to randomisation (with wine drinkers seeing one of the 18 wine bottles, and beer drinkers one of the 18 beer bottles). Then participants rated on a scale from 'Definitely Not' to 'Definitely Yes' whether the presented drink was likely to appeal to each of the different target groups and target occasions.
- Indeed, both the PCA and the descriptive analyses compared all 18 drink labels, including contrasting the labels with both the verbal descriptor and %ABV, and those with just the verbal descriptors. This can be most clearly seen visually in Figures 2 and 3.

Was participants' own drinking assessed to test whether own drinking was associated with their perceptions? Were participants' descriptive or injunctive norms assessed, in particular, what occasions they think others drink or that they find acceptable to drink any alcohol, regardless of strength? Did perceptions differ as a function of participant age or gender?

- Regarding participant-level measures:
 - We measured participants' level of drinking using the AUDIT-C as a simple descriptive indicator of individual-level differences between participants (see Table 1).
 - We did not assess participants' descriptive or injunctive norms.
- Regarding moderated analyses:
 - The analyses you suggest with the variables age, gender and self-assessed drinking level are interesting. We tried to perform them (by means of likelihood ratio tests comparing logistic regression models explaining the probability of finding a drink appealing for a given group or occasion as a function of our 18 descriptors with and without moderation of the variable of interest) and decided not to report these analyses as they suffer from lack of statistical power (as suggested by the large confidence bounds noted in Figures 2 and 3), partly due to the between-subjects design: after multiplicity correction, no effect could be detected.
 - We also investigated to possibility to add a predictor to a PCA, but, to the best of our knowledge, such analyses do not seem to exist.

The authors seem to argue that offering a low strength alcohol would increase overall alcohol consumption because they found that participants reported that the substance might appeal to a wider variety of people and that others might drink this lower strength alcohol in more situations than regular strength alcohol use. However, the results do not suggest that more categories of people will drink this beverage and on more occasions than regular strength alcohol. The lack of behavioral outcome data makes this conclusion an extrapolation that is not necessarily supported by data in this study. There is no evidence from this study that people would drink more of the lower strength alcohol or that by potentially increasing drinking occasions, that problems would result, so these potentialities should be walked back some throughout the paper.

- We disagree with this comment. Our analysis shows that a large sample of weekly wine and beer drinkers perceives lower strength wines and beers as being more likely to appeal to a wider variety of people (such as pregnant women, dieters etc.), and also more likely to be appealing for non-traditional occasions such as weekday lunchtimes, when compared to perceptions of the target groups and occasions of Regular labelled wine and beer. Hence, our conclusions hold-up to the data. Our discussion of the pattern of results also uses language to appropriately highlight that these results are suggestive of the potential for lower strength alcohols to increase both the type of groups and the type of occasions when such alcohol is consumed. Importantly, we discuss this pattern of results in relation to extant literature on this topic, which also recently showed that UK retailers and producers market lower alcohol drinks as more appropriate for drinking by more people during more occasions when compared to the marketing of regular alcohol drinks. We also discuss a recent study which found in a bar laboratory setting that participants drink more alcohols that are labelled as lower in strength (when the only thing that changes is the label, but not the actual drink). We do discuss at length that we do not have behavioural data, and we discuss how this limitation fits with the findings of the present study.

Throughout the study the authors use language that participants were rating acceptability, suitability, or approval of drinking the lower strength alcohol by specific subgroups of people and on particular occasions. However, the participants were asked how likely those people would drink lower strength alcohol and what occasions people would likely consume the lower strength alcohol. Likelihood is not the same as approval or suitability. The participants were not necessarily conveying that they approved of these individuals and occasion for drinking low strength alcohol. That distinction is very important and the paper should be revised to reflect that distinction. For example, the abstract says "...the kind of person who would find the beverage most appealing, and the types of occasions on which the beverage could be drunk", to more accurately reflect the findings, the authors could change this to "subtypes of people they perceived to be more likely to drink the lower strength alcohol and occasions at which people would be more likely to drink it." Or something to that effect.

- We thank you for your suggestion. We have made our language more consistent throughout the revised manuscript.

Minor Points:

Page 8 line 45 and 46 appear larger than the text in the rest of the paragraph.

- Thanks for drawing our attention to this. We have reformatted these particular lines.

Page 9 line 52 insert and between drivers and sportspeople

- Thanks, we have inserted "and" between drivers and sportspeople.

Page 13 line 3 underage should be one word

- We disagree with this suggestion. Under age is the correct phrase to use in this particular sentence (see the Collins English Dictionary: <https://www.collinsdictionary.com/dictionary/english/under-age>).
- We are also providing the sentence under discussion in its entirety here:

<< Analyses of the perceived target groups for consumption showed that participants perceived 0% and other lower strength labelled drinks as targeting pregnant women, dieters, drivers, sportspeople, and those under age. In terms of targeted occasions, 0% and other lower strength drinks were perceived as targeting weekday lunchtimes. >>

The font in the tables is not times new roman

- BMJ Open does not follow APA style, hence the font of the tables is not Times New Roman.

On page 6 line 39, APA style does not allow for numbers at the beginning of sentences.

- BMJ Open does not follow APA style. We have nevertheless made the suggested edit.

Add a comma after the work link on page 6 line 43.

- This comma has been added now.